# See and Copy: Generation of complex compositional movements from modular and geometric RNN representations

**Sunny Duan**[†]                                                           sunnyd@mit.edu
**Mikail Khona**[†]                                                           mikail@mit.edu
**Adrian Bertagnoli**[†]                                                      ab_@mit.edu
**Sarthak Chandra**                                                        sarthakc@mit.edu
**Ila Fiete**                                                                 fiete@mit.edu

[†]**denotes co-first authors**

**Editors:** Sophia Sanborn, Christian Shewmake, Simone Azeglio, Arianna Di Bernardo, Nina Miolane

## Abstract

A hallmark of biological intelligence and control is combinatorial generalization: animals are able to learn various things, then piece them together in new combinations to produce appropriate outputs for new tasks. Inspired by the ability of primates to readily imitate seen movement sequences, we present a model of motor control using a realistic model of arm dynamics, tasked with imitating a guide that makes arbitrary two-segment drawings. We hypothesize that modular organization is one of the keys to such flexible and generalizable control. We construct a modular control model consisting of separate encoding and motor RNNs and a scheduler, which we train end-to-end on the task. We show that the modular structure allows the model to generalize not only to unseen two-segment trajectories, but to new drawings consisting of many more segments than it was trained on, and also allows for rapid adaptation to perturbations. Finally, our model recapitulates experimental observations of the preparatory and execution-related processes unfolding during motor control, providing a normative explanation for functional segregation of preparatory and execution-related activity within the motor cortex.

**Keywords:** Modularity, Motor Control, Neural Representational Geometry

## 1. Introduction

Animal behavior is believed to be atomic: composed of a discrete set of "syllables" or motifs which together form a "grammar" Wiltschko et al. (2015); Markowitz et al. (2018). Action sequences are then generated by combining syllables and executing them sequentially. The compositional structure inherent in this scheme allows animals to flexibly recombine motor primitives to generate novel, ecologically relevant movement patterns. This system affords animals a combinatorially large movement repertoire built of out simpler components. However, the specialized structures in the motor system required to implement this scheme efficiently remain unknown.

Existing experiments in the motor cortices of non-human primates performing reaching movements have indicated that the process of movement generation is comprised of three interdependent stages: a preparatory stage, a trigger signal and action execution. During the preparatory stage, the parameters for a motion are able to be decoded from the motor cortex Churchland et al. (2010) indicating that the planned trajectory is present in the

Duan† Khona† Bertagnoli† Chandra Fiete

motor cortex despite the lack of motion production. After the onset of the trigger signal, the motor cortex exhibits dynamics which produce downstream muscle activations leading to the desired motion.

In order to combine multiple motor primitives into complex sequences, there must be additional structure to support these longer sequences. Recent work studying the neural dynamics of rhesus macaques performing skilled, practiced compound reaches has indicated that conjunctive movement consists of separate independent chains of motor processes sharing the same underlying neural substrate Zimnik and Churchland (2021). In order to skillfully and smoothly execute a sequence of movements, the motor cortex needs to simultaneously prepare for an upcoming movement while current motor commands are being executed. This multi-tasking capability requires functional segregation preparatory activity and execution within the underlying motor cortical circuitry. Existing research suggests that this is implemented in biological systems by separating preparatory activity and execution into orthogonal subspaces.

We demonstrate that a task-optimized neural network is able to implement arbitrary motor subroutines in a flexible and reusable manner. Our model demonstrates remarkable generalization capabilities due to its structure. Through learning, our model exhibits emergent self-organization of its latent representation which facilitates robust production of movement patterns. Furthermore, our model is able to continuously produce sequences of motion interference, demonstrating properties of functional segregation exhibited in biological neural circuits.

## 2. Methods

Our model architecture resembles the sequence-to-sequence models used in natural language processing Sutskever et al. (2014). The input data (from the observed "guide") consists of procedurally generated movements consisting of up to two straight segments, represented by a sequence of x,y coordinates. The agent constructs an embedding by sequentially ingesting the x,y coordinates of the guide sequence into an encoder implemented as a recurrent neural network (RNN) consisting of continuous-time neurons which we denote as continuous time recurrent neural networks (CT-RNN) Appendix A.1. The encoder produces a single (static) readout, or embedding, at the end of each segment. These embeddings are fed into the motor RNN at pre-specified times by a scheduler, which modulates the embedding by a ramping function such that motion onset is triggered by the falling edge of the ramping signal which is analogous to a 'go cue' used in animal experiments Hennequin et al. (2014). The ramping signal is multiplied by the embedding of the subsequently executed segment such that the motor RNN does not receive information about the movement until the ramping signal has commenced. The beginning of the ramp signal was timed such that the embedding signal was provided 7 steps before the end of the previous segment such that the go signal aligns with the end of the previous segment. This setup effectively 'loads' the embedding of the movement into the motor RNN prior to its execution. However due to the continuous nature of the compound movement, the temporal overlap of the preparation and execution phases force the network to simultaneously process both the future motor command while completing the current segment.

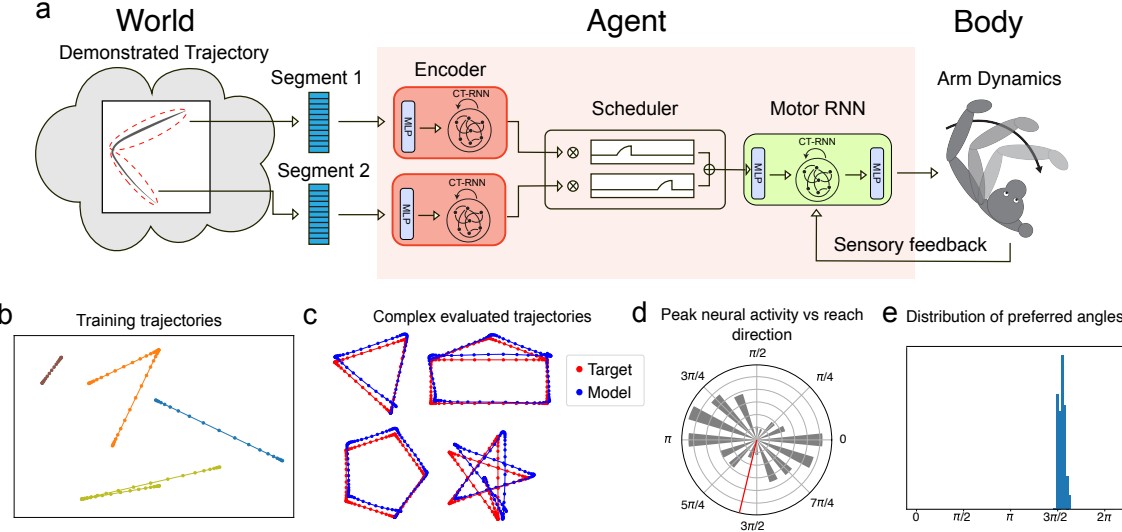

Figure 1: a) Schematic of encoder-decoder training setup with trained continuous time recurrent neural networks (CT-RNN) in both the Encoder and Motor RNN components of our system. b) Example trajectories used for training the model, consisting of procedurally generated single and 2-segment trajectories of various length and angles. c) Examples of complex, out-of-distribution sequences generated by the trained model. d) Polar histogram indicating the distribution of preferential movement direction. Each neuron is assigned a preferential movement direction based on its activity profile. The red line indicates the optimal movement direction e) Distribution of preferred reach direction for different starting locations within the drawing board, computed by finding the reach direction which minimized error.

DUAN† KHONA† BERTAGNOLI† CHANDRA FIETE

The motor RNN takes in the ramp-modulated static encoder embedding and produces a sequence of muscle activation commands which are implemented by a realistic simulation of an over-actuated two-link arm moving in two dimensional space Kalidindi et al. (2021), Lillicrap and Scott (2013). Our model receives feedback about angular position and velocity of the joints of the arm but must learn to time the muscle activity in order to produce the target motion sequence. The entire system is trained end-to-end on one or two segment motions using stochastic gradient descent and supervised learning using mean squared error of the virtual pen with respect to the target sequence at each moment in time. Our model additionally included biologically relevant loss terms penalizing neural activity and simultaneous activation of antagonistic muscle pairs. Appendix A.3

## 3. Results

Although our model was trained on sequences of straight lines with at most two segments (Figure 1b), we find that our network is able to generalize to sequences of arbitrarily many more segments, enabling the network to draw complex figures which it has never been trained on before (Figure 1c). We posit that this ability arises from the implicit modular structure of our network. This modularity is consistent with recordings from the primate motor cortical system Zimnik and Churchland (2021). We find that the architectural segmentation of movement production into a preparatory phase and execution-related phase, motivated by electrophysiology data, encouraged the network to produce relaxation dynamics and permitted it to reuse existing circuitry for the production of conjunctive sequences. This is a result of the decoder's ability to cycle through phases of preparation and execution in which the clear separation of stages and the tendency of the dynamics to return to a fixed point allows the activity to 'reset' between movements and prevents interference from prior movements. We also demonstrate that the structural priors imposed on our model give rise to characteristic behaviors of biological motor control including orthogonality of preparatory and execution-related activity and rapid adaptation to changes in physics.

### 3.1. Preferred movement directions

We analyzed the preferred movement direction for each neuron. This was done by quantizing the possible movement directions and determining the direction which caused maximal activation for each neuron, which indicates its preferred movement direction. Here we found a bimodal distribution with peaks located approximately 180 degrees from one another Figure 1d. This phenomenon has been previously reported by Lillicrap and Scott (2013), in which this effect is attributed as the optimal neural activity for the biomechanical properties of the limb. Next, we investigated whether the preferred movement directions would allow us to predict the optimal movement direction, which we defined as the direction which minimized the cumulative squared error for a short trajectory. As shown in Figure 1e, this direction was tightly distributed around 270 degrees. In the specific starting position used to generate Figure 1d, the optimal direction was at 261 degrees, which is nearly perpendicular to the peaks of the bimodal distribution of preferred directions.

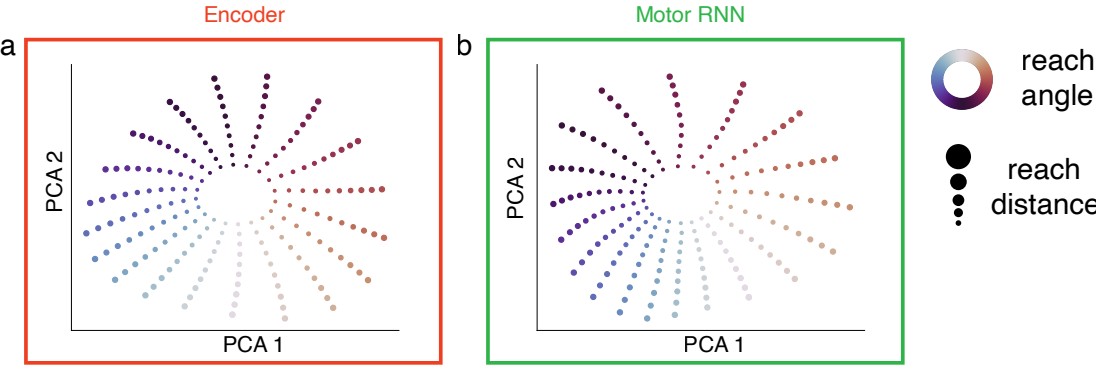

Figure 2: a) PCA embedding of the encoder latent state which is fed into the motor RNN over a dataset of single-segment reaches starting from the same initial position, colored by reach angle and with size corresponding to the reach distance. This embedding represents the input that the motor RNN receives during the preparatory phase. b) PCA embedding of the motor RNN latent state one step before motion onset, similarly colored and sized. The manifold of embeddings are homeomorphic to a cylinder and is parameterized by reach angle and distance. The space of the encoded latent state and motor RNN latent state right before motion are almost identical up to a linear transformation, indicating that the motor RNN effectively loads in the representation produced by the encoder.

### 3.2. Structured latent representations of movement

We investigated how guide trajectories were represented in our motor RNN. We instructed our model to hold for a fixed period during which the embedding of the trajectory was fed into the decoder, emulating experiments on non-human primates Churchland et al. (2010). In our experiments, we are able to decode future movement from the motor RNN one time-step prior to motion onset with an mean average error of 0.129 radians on a held-out dataset indicating that the relevant movement parameters were present in the motor RNN state. We also observed that low-dimensional PCA projections of neural activity during reaches of varying angle and length exhibit rich topological structure. Visualizing the projections onto the top 2 principle components show that the manifold of latent representations in both the encoder and the motor RNN are homeomorphic to a cylinder parameterized by the angle and length of the reach (Figure 2). This smooth mapping of movement parameters presumably allows the network to robustly encode the desired trajectory and provides an initialization of the motor RNN state which evolves autonomously.

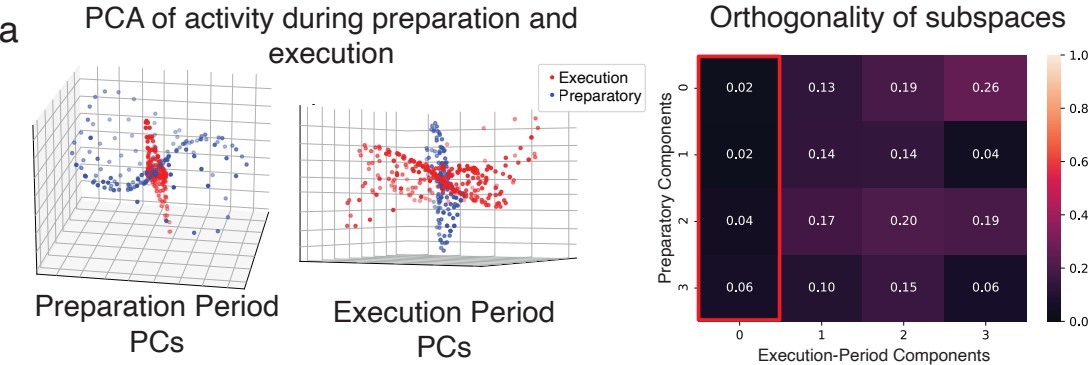

Figure 3: a) Visualization of the the neural activity during the preparatory period (blue) and execution period (red) projected onto the top 3 principle components of the neural activations during preparation period (left) and the same neural activity projected onto the top 3 principle components of the neural activity during execution (right). b) Computed subspaces consisting of the first 4 principle components of the neural activity during the preparatory period and execution period. We quantify the orthogonality of the subspaces by taking the dot products of the PCs. We observe that the top PC of the execution-period components are completely orthogonal to all of the components of the preparation period components.

### 3.3. Orthogonality of neural activity during preparation and execution

Additionally we sought to understand how our network was able to simultaneously execute an existing motor command while reading in future commands without interference. In line with experimental results, we observed that our network separated preparatory activity and execution into orthogonal subspaces Churchland et al. (2010). Visual inspection of

the principle components of preparatory activity and execution-related activity reveals this relationship (Figure 3a). Quantifying this relationship by computing the top 4 principle components, which account for 97.7% and 96.7% of the preparatory and execution-related activity respectively, shows that these components are largely orthogonal and the top principle component of the execution period is orthogonal to all of the components of the activity in the preparatory period (Figure 3b). The subspaces which contain the activity during preparatory phase and execution phase do overlap, however, the direction which accounts for the maximum amount of variance during the execution phase is completely absent during preparation. Qualitatively, we observe that during the preparation phase, the neural activity spreads out along a hyperplane and this entire space rotates onto a new set of axes during the execution phase, during which the activity contracts back to the origin (Figure 3a). We posit that orthogonality of these processes emerges as a consequence of the need for the circuit to support both preparatory and execution-related processes simultaneously.

### 3.4. Embedding reuse in multi-segment sequences

We compared movements consisting of two segments with single segment movements to evaluate whether the network reused the same neural activity for both the multi-segment and the single-segment movement. Our model was commanded to perform randomly sampled 2-segment movements and then, independently, only the second half of the 2-segment movements. Computing PCA on all of the neural activity showed that there were 6 principle components accounting for 92% of the variance. Visualizing the evolution of these components over time reveal that the second half of the 2-segment movements quickly converged to the same trajectories as the 1-segment movements even in cases when the activity had not returned to zero yet (Figure 4). This demonstrates that our network not only reuses the same neural representation for conjunctive movements but also is able to avoid interference by neural activity from the first segment.

### 3.5. Rapid adaptation to changes in arm physics

Many experimental studies have used perturbations of underlying physics of the task in order to assess the role of the motor cortex (Cherian et al. (2013); Vyas et al. (2018); Sun et al. (2022)). We modeled this by assessing our models ability to recover from changes in biomechanical arm parameters. After increasing the moment of inertia of our simulated arm by 30%, we observed significant position errors. However, after less than 20 additional gradient steps confined to the weights of only the motor RNN (keeping the encoder RNNs fixed), our model was able to rapidly adapt to the change in inertia to recover its original performance, assessed on a held out evaluation set of 40 movements. (Figure 5a)

### 3.6. Feedback knockout

The dynamics of the motor network are driven by a combination of internal recurrence, and time-varying feedback from the arm. We sought to characterize how much of our dynamics were derived from autonomous temporal dynamics in the motor network based on the static information contained in the preparatory state set by the encoding network output. We knocked out feedback from the arm and observed how its performance degraded while repeatedly drawing a diamond. The knockout model still produced approximately the

DUAN[†] KHONA[†] BERTAGNOLI[†] CHANDRA FIETE

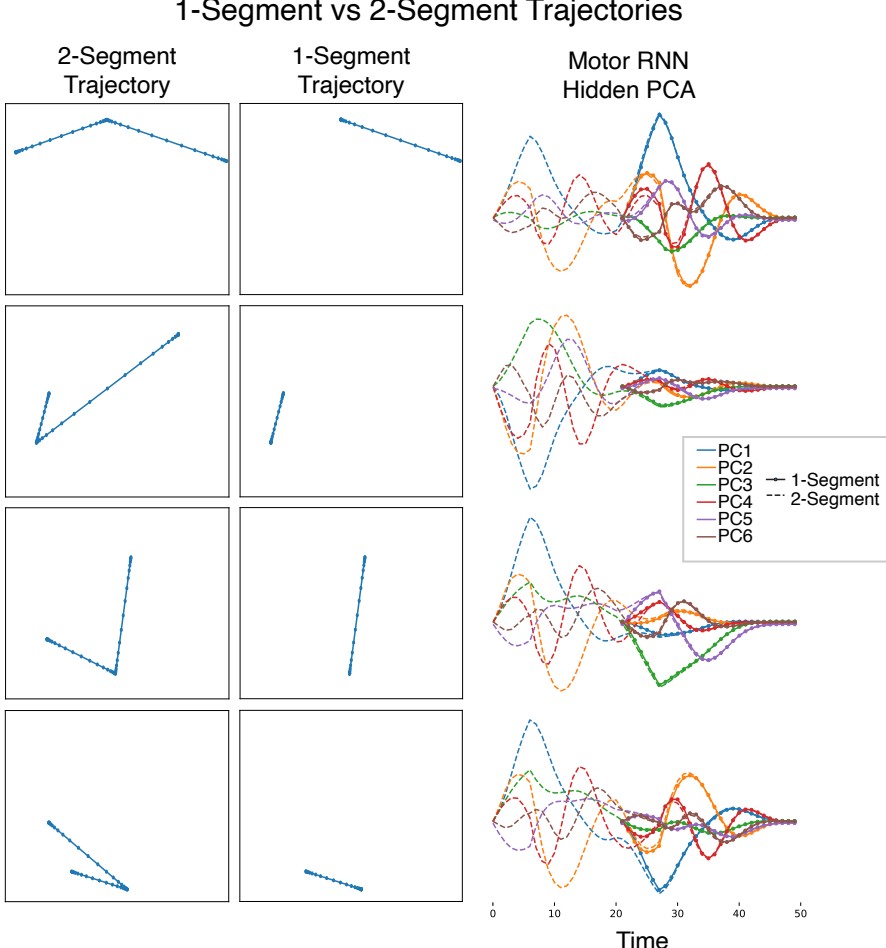

Figure 4: Comparison of neural activity for 2-segment trajectories and 1-segment trajectories consisting of the second half of the 2-segment trajectories. In the right figure we've plotted the first 6 PCs, which account for 92% of the variance and the neural activity. The 1-segment trajectories are plotted in solid lines, and 2-segment trajectories are plotted in dotted lines. Notably, the second half of the 2-segment trajectories converges almost identically to the corresponding 1-segment trajectory despite not yet returning fully to zero before the onset of the second segment. This implies that the motor RNN executes 2-segment sequences as the conjunction of two single segment sequences and demonstrates reuse of the neural circuit despite the additional separating work the network must do in the two-segment trajectories to deal with the initial overlap of the two segments.

correct shape, but gradually drifted away from the target location, indicating that the model utilized feedback mostly as a correctional signal rather than a major source of dynamical drive.

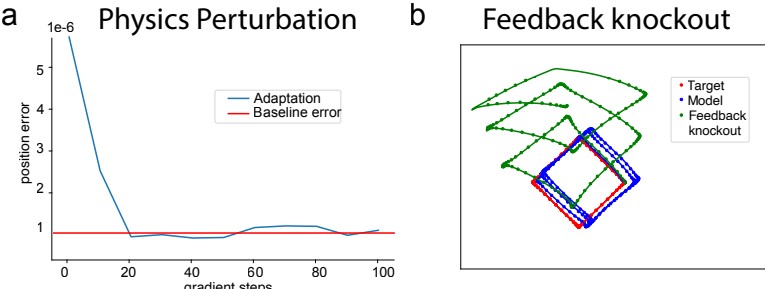

Figure 5: a) Performance of our model while recovering from changing the arm moment of inertia by 30%, training only the motor RNN portion of the model. b) Comparison of our model tracing a diamond multiple times with and without feedback.

## 4. Discussion

Our model takes inspiration from the versatility of animals in producing novel movement outputs and of primates in their ability to "mirror" seen movements to generate combinatorially flexible movements with a biophysical multi-jointed model arm. Although it is difficult to make claims about how faithful RNNs are as a model of biology, our model provides confirmation that observed phenomena of motor cortical dynamics have functional utility in allowing neural circuits to reuse previously seen motor primitives and recombine them into complex sequences of movements. We found that the orthogonality of preparatory and execution-related activity emerged in our end-to-end trained model from the need to separate incoming planned movement instructions from generating ongoing movements and is indicative that this phenomenon is needed in order to prevent interference between subsequent movements. Additionally, the separation of movement production into preparatory and execution phases allowed our model to robustly generalize to significantly more complicated movements. We find that both of these phenomena are critical to facilitating modularity and generalization in RNNs and our experiments provide insight into why these structures may exist in biology.

In our study, we focused on how the motor RNN is able to learn and composibly reuse movements. However, a complete model of compositional control would require the ability to decompose arbitrary drawings into modular sub-routines, which, in our study was supplied by an external trigger signal. Additionally, we focused on a limited set of movements consisting of combinations of straight reaches whereas realistic drawing require a larger repertoire of primitive movements. These are natural directions for future study.

The ability of an agent to decompose complex tasks into subroutines is a core component in how intelligent agents are able to adapt to new environments and form complex behaviors.

This skill acquisition capability has been studied in the context of reinforcement learning in the hopes of producing artificial agents which are able to solve novel tasks Sharma et al. (2019) Xu et al. (2020). Biological systems are able to efficiently acquire new skills and recombine them in novel environments, solving tasks which are inaccessible to current state of the art artificially intelligence systems. These systems have been evolutionarily optimized to efficiently learn and perform tasks that are critical for survival; thus, they provide clues for how to engineer artificially intelligent systems to exhibit similarly remarkable capabilities.

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

## Appendix A. Model Architecture

There are two neural network models in our model: The encoder model and the motor RNN. They both contain a continuous time recurrent neural network which is a variant of the traditional recurrent neural network.

### A.1. Continuous Time Recurrent Neural Network (CT-RNN)

The CT-RNN component of our network is a continuous time RNN with tanh non-linearity, discretized using the forward eurler method as a discrete approximation for the continuous time dynamics.

$$\tau \frac{h[t+1] - h[t]}{\Delta t} = \sigma(x + Wh[t])$$

where $h[t], h[t+1]$ represents the vector-valued state of the hidden units at time step $t$, $W$ is the recurrent weight matrix, $x$ is the input from upstream layers, $\sigma$ is the tanh non-linearity and $dt$ and $\tau$ are constants with the values $dt = 1.0$ and $\tau = 100$. In our network the hidden layer size is 1024 units.

### A.2. Input/Output Layers

In addition to the CT-RNN module, our encoder is furnished with a 2-layer multi-layer perceptron (MLP) which takes in the x,y coordinates as input and has sizes $2 \times 1024$ and $1024 \times 1024$ with an intermediate ReLU non-linearity. The output of this network is fed in to the CTRNN as input.

The decoder also has an input MLP which takes in as input the concatenation of the 1028 unit embedding produced by the encoder along with the x,y coordinates of the current tip location, the two angles specifying the position of the arm joints, as well as the current angular velocity of the two arm joints for a full input size of 1034. This is fed into a 2-layer MLP with sizes $1034 \times 1024$ and $1024 \times 1024$ with a ReLU as the intermediate non-linearity. The decoder is also furnished with a 2-layer MLP with sizes $1024 \times 1024$ and $1024 \times 6$ which produces a 6 dimensional muscle activation vector which is consumed by our realistic arm environment.

### A.3. Loss Function

Our model was trained to reproduce a target sequence of points using mean squared error with additional terms penalizing hidden neuron activation and antagonistic muscle activation. Thus our loss has the following form:

$$\mathcal{L}(y, \hat{y}, h, u) = \frac{1}{L}\|y - \hat{y}\|_2^2 + \alpha\|h\|_1 + \beta(u_{\text{extend}} \cdot u_{\text{contract}})$$

Where $y$ and $\hat{y}$ are the respective target sequence and model-produced sequence of x,y positions of the pen tip, $h$ is the vector of hidden activations from the motor RNN, $u_{\text{extend}}$

and $u_{\text{contract}}$ are the produced muscle activation vectors for the muscles which extend the arm joints and the corresponding contracting muscles, $L$ is the sequence length, $\alpha, \beta$ are weighting terms and $\|\cdot\|_2, \|\cdot\|_1$ are L2 and L1 norms respectively.

