# OpenReview forum: "See and Copy: Generation of complex compositional movements from modular and geometric RNN representations"
_NeurIPS.cc/2022/Workshop/NeurReps — NeurReps 2022 Poster_

### Official Review · Reviewer_3jLm · 2022-10-14
**Representation learning in control signal space is an important direction.**

**Confidence:** 3
**Soundness:** 3
**Presentation:** 2
**Contribution:** 3
**Overall Rating:** 7

**Summary:**

Overall, I think representation learning in control signal space is an important direction, which did not receive enough attention historically, possibly because the systems to be controlled are typically designed rather than given by nature. Or in other words, we start with a rough representation, if not a precise one. Given the importance of the topic and a lack of effort in this general direction, I recommend accepting this paper.

However, I feel that the authors may want to make a reasonable effort to improve the presentation, and the current manuscript may not deliver the message well.

1. What is the scheduler doing? According to Figure 1(a), it looks like a temporal vector signal with two pulse inputs to the Motor RNN. However, when the motor command is executed frequently, the motor signal shall be persistent rather than a pulse. So this formulation looks counterintuitive. Further, why do we need this temporal input during training? The two-pulses construction feels redundant.

2. Still, in Figure 1, does CT-RNN refer to continuous time RNN? If so, it might be good to state that clearly, as section 2 only says, "RNN consisting of continuous-time neurons .."

3. The authors may want to explain more about figure 1(d), what is the intuition for it?

4. Figure 2 (a) and Figure2 (b) look very similar, what should a reader learn from this figure?

5. In the Figure 3 caption, the authors state, "we observe that the top PC of the execution-period components are completely orthogonal to all of the components of the preparation period components." What is the embedding dimension? If the embedding dimensional is a high-dimensional space, then isn't this expected? The authors shall provide more details and intuition to guide the readers to learn better from this phenomenon.

6. In Figure 3(a), the two subfigures are very similar, except for the opposite of color. Please provide more guidance.

7. In Figure 4, the motor RNN Hidden PCA part can be made more clear. It's a very intuitive concept, and the visualization shall be made better.

The comments and questions are merely to help the authors improve the manuscript and polish the message. And I like the topic, and the results are helpful. Another suggestion is that the authors may want to discuss the geometry of the representation more.


**Questions:**

Please refer to the Summary section.

**Limitations:**

Please refer to the Summary section.

**Recommended Decision:**

3: Accept

**Relevance:**

2: Limited relevance

**Strengths And Weaknesses:**

Please refer to the Summary section.

**Submission Track:**

Proceedings Paper (9 Page)

---

### Official Review · Reviewer_Wu8G · 2022-10-18

**Confidence:** 1
**Soundness:** 3
**Presentation:** 3
**Contribution:** 3
**Overall Rating:** 6

**Summary:**

The authors present a model of motor control using two RNNs which allows to learn how to move an arm. They also provide an explanation of what is happening inside the motor cortex.

**Questions:**

Citations should be in parenthesis

**Limitations:**

It would be interesting to try other models

**Recommended Decision:**

2: Borderline

**Relevance:**

4: Highly relevant

**Strengths And Weaknesses:**

Strenghts:
- Interesting problem
- Principled method
- Clear analysis

**Submission Track:**

Proceedings Paper (9 Page)

---

### Official Review · Reviewer_1qRL · 2022-10-18
**Generation of complex compositional  movements from modular and geometrical RNNs representations**

**Confidence:** 5
**Soundness:** 4
**Presentation:** 3
**Contribution:** 4
**Overall Rating:** 8

**Summary:**

This paper presents a model of the learning and control of sequential movements made by a primate or human-like arm based on a modular structure composed of two RNNs and a scheduler. The model examined the ability of the suggested model to learn drawing movements composed of two segments and to generalize its learning to the drawing of more complicated drawings composed of a larger number of straight lines.
The results are examined with respect to a recent paper describing the neural activities of neurons recorded from monkeys' brains during the performance of sequential tasks. In particular,  the results recapitulate the experimental observations showing a modular organization and the decomposition of the neural populations into sub-populations engaged in either movement preparation or motor execution processes.

**Questions:**

There are several issues that are not clear.  The work uses a computational; model of the arm acted upon by muscles.  Does the model include also inverse and forward dynamic models? also not clear is what cost the optimization involves,  It is stated that it penalizes
the simultaneous activities of antagonistic muscles and neural activation but it does not explain what it uses such a loss function nor does it include what objectives are optimized-, e.g., does it include an end-point accuracy cost or other kinematic costs?
Also the ramping activity of the scheduler -is it based on experimental data derived from monkey studies performing similar tasks?
Also,  there is not a sufficiently clear explanation of the results shown in figure 2.  What do the 4 PCA represent?
Also, are the temporal patterns of muscles' activities explicitly represented? Does the orthogonality between motion preparation and execution result from the two-burst patterns of agonists versus antagonistic contractions? or, please explain better what this result can be attributed to. It is also not clear whether the network satisfactorily predicts kinematic profiles of the movements and not only the paths'  deviations from straight lines because of the influence of the arm inertial characteristics.

**Limitations:**

The authors should explain what are the limitations of the study.


**Recommended Decision:**

3: Accept

**Relevance:**

4: Highly relevant

**Strengths And Weaknesses:**

Original: the work is quite interesting and novel and is closely linked to motor neurophysiological studies, providing a nice computational explanation for the observed modularity of neural populations.  Quality: the work is significant and illustrates the usefulness of using neural networks-based research to explain neurophysiological findings and behaviorally observed motion characteristics.
Clarity: the manuscript is clearly written and explained and except for minor points that are missing the approach used and the tests conducted are well explained and convincing.

**Submission Track:**

Extended Abstract (4 Page)

---

### Decision · Program_Chairs · 2022-10-21

Accept (Poster)